# Ambulatory management of pre- and extensively drug resistant tuberculosis patients with imipenem delivered through port-a-cath: A mixed methods study on treatment outcomes and challenges

**Vijay Vinayak Chavan**[1]*, **Alpa Dalal**[2], **Sharath Nagaraja**[3], **Pruthu Thekkur**[4,5], **Homa Mansoor**[1], **Augusto Meneguim**[1], **Roma Paryani**[1], **Pramila Singh**[1], **Stobdan Kalon**[1], **Mrinalini Das**[1], **Gabriella Ferlazzo**[6], **Petros Isaakidis**[6]

1 Médecins Sans Frontières/Doctors Without Borders, Mumbai, India, 2 Jupiter Hospital, Thane, Mumbai, India, 3 ESIC Medical College and PGIMSR, Bengaluru, India, 4 International Union Against Tuberculosis and Lung Diseases, Paris, France, 5 The Union South East Asia Office, New Delhi, India, 6 Southern Africa Medical Unit, Médecins Sans Frontières, Cape Town, South Africa

* dr.vijayvchavan@gmail.com

**Data Availability Statement:** Data is coded and stored at MSF independent clinic .Mumbai. since

## Abstract

### Background

Imipenem, an intravenous antibiotic is recommended for use in drug resistant tuberculosis (DR-TB) when an effective regimen with combination of other second line drugs is not possible. Though the treatment success rates with carbapenems are promising, the twice daily injection of Imipenem usually requires patients to be hospitalized. The Médecins Sans Frontières independent clinic in Mumbai, India implemented ambulatory and home based management of patients receiving Imipenem through the use of port-a-cath.

### Objective

We aimed to describe the adverse events and treatment outcomes of ambulatory pre- and XDR-TB patients initiated on imipenem through port-a-cath between January 2015 and June 2018 and to explore the challenges with this regimen as perceived by healthcare providers and patients.

### Methods

A convergent mixed methods study with quantitative (longitudinal descriptive study using the routine data) and qualitative (descriptive study) part conducted concurrently. For the quantitative component, all XDR-TB and pre-XDR-TB initiated on imipenem containing regimen during January 2015-June 2018 were included. For qualitative component, interviews were carried out including patients who initiated on imipenem (n = 5) and healthcare providers (n = 7) involved in providing treatment. Treatment outcomes, culture conversion and

the data is of patients with personal information, data cannot be shared publically due to patient confidentiality issues. Data are available on request in accordance with MSF's data sharing policy. Requests for access to data should be made to data.sharing@msf.org For more information please see: 1) MSF's Data Sharing Policy: http://fieldresearch.msf.org/msf/handle/10144/306501 2) MSF's Data Sharing Policy PLOS Medicine article: http://journals.plos.org/plosmedicine/article?id=10.1371/journal.pmed.1001562.

**Funding:** The authors received no specific funding for this work.

**Competing interests:** The authors have declared that no competing interests exist.

adverse events during treatment were described. Thematic analysis was carried out for qualitative component.

## Results

Of the 70 patients included, the mean age was 28.1 (standard deviation: 11.2) years and 36 (51.4%) were females. Fifty one (72.9%) had XDR-TB. All patients were resistant to fluoroquinilone, levofloxacin. Vomiting was reported by 55 (78.6%) patients and at least one episode of QTC prolongation (more than 500 msec by Fredrecia method) was detected in 25 (35.7%). Port-a-cath block and infection was seen in 11 (15.7%) and 20 (28.6%) patients respectively. Favourable outcomes were seen in 43 (61.4%) patients. Mortality was seen in 22 (31.4%) patients, 2 (2.9%) were lost-to-follow-up and 3 (4.3%) were declared as treatment failure. The overarching theme of the qualitative analysis was: Challenges in delivering Imipenem via port-a-cath device in ambulatory care. Major challenges identified were difficulties in adhering to drug dose timelines, vomiting, restricted mobility due to port-a-cath, paucity of infection control and space constraints at patients' home for optimal care.

## Conclusion

Administration of imipenem was feasible through port-a-cath. Though outcomes with ambulatory based imipenem containing regimens were promising, there were several challenges in providing care. The feasibility of infusion at day care facilities needs to explored to overcome challenges in infusion at patients home.

## Introduction

Multidrug resistant tuberculosis (MDR-TB) and extensively drug-resistant tuberculosis (XDR-TB) pose challenges for global TB control efforts with low treatment success rates and high rate of death [1]. Globally, among patients initiated on treatment in 2016, only 56% and 39% of the MDR-TB and XDR-TB patients had successful treatment outcomes [2].

In order to achieve the target of "zero TB mortality" by 2030, there is need to address the poor treatment success rates among drug resistant TB patients (DR-TB) [3]. However, the standardised regimens for DR-TB are shown to be ineffective in a considerable proportion of cases due to safety profile of the drugs and to emergence of varied resistance to second line TB drugs [4,5].

Considering the complexities involved in treating DR-TB, WHO recommended for designing customized drug regimens [6]. To facilitate this, the drug basket for management of DR-TB was widened with inclusion of newer drugs. The choice of drugs in the individualized drug regimens are made based on a preference for oral over injectable agents; the results of drug-susceptibility testing (DST); the reliability of existing DST methods; population drug resistance patterns; history of previous drug exposure in a patient; drug tolerability; and potential drug-drug interactions [7]. However, the most recent guidelines for management of DR-TB suggest for customizing all oral shorter regimens with newer drugs like bedaquiline, pretomanid and linezolid [1].

Imipenem containing regimen is recommended if an effective regimen with second line drugs cannot be built due to resistance or intolerance. Studies among XDR-TB patients receiving imipenem containing regimens have shown relatively high sputum conversion (72%) and

treatment success rates (60%) [8,9]. Though the success rates are encouraging, the regimen requires multiple intravenous injections and hence, the hospitalization for treatment [10].

The Médecins Sans Frontières (MSF) independent clinic in Mumbai, India with expertise in managing patients on individualised treatment regimens developed ambulatory management of XDR and pre XDR-TB patients with imipenem containing regimen. The clinic adopted the technique of administering imipenem through port-a-cath, a small device placed under the skin on the right side of the chest. It is attached to a catheter (a thin, flexible tube) that is threaded into a large vein (superior vena cava) [11]. Though port-a-cath is used in medical conditions requiring multiple intravenous injections to administer fluids, blood, chemotherapy and other drugs, its use in management of DR-TB is not documented globally. The port-a-cath stays in place for weeks or months and helps to avoid repeated needle pricks to the patients. Thus, it allows patients to receive ambulatory care which is the preferred model of DR-TB care [12].

Though the MSF clinic is treating patients with imipenem through port-a-cath since 2015, there has been no reporting of systematic assessment of adverse events related to port-a-cath and treatment outcomes among these patients. Evidence on safety and efficacy of treatment regimens including imipenem administered through a port-a-cath device is needed to guide policy and clinical practise. Documentation of the challenges experienced by health care providers and patients is an important area to be explored for optimizing the management.

This is a mixed methods study with specific objectives; 1) to describe the adverse events and treatment outcomes of XDR-TB and pre-XDR TB patients initiated on imipenem containing regimen between January 2015 and June 2018 at MSF independent clinic in Mumbai and 2) to explore the challenges associated with imipenem containing regimens as perceived by healthcare providers and patients.

## Methods

### Study design

This was a convergent mixed methods study with the quantitative part (longitudinal descriptive study using the routinely collected data by the MSF independent clinic) and the qualitative part (descriptive study) conducted concurrently [13].

### Study setting

**General setting.** Mumbai is a metropolitan city in the West coast of India. The city is densely (~73,000 per square miles) populated with an estimated population of over 22 million. The city has highest number of drug resistant TB patients in the country with around 5000 MDR-TB and 700 XDR-TB patients. The free care to TB patients is provided in all public health facilities under National TB programme and select clinics run by international NGOs like MSF. The private healthcare providers also treat and notify DR-TB patients.

**Specific setting.** The study was conducted in the MSF independent clinic situated in Mumbai specialised in providing care to DR-TB patients. The clinic has a multi-disciplinary team of doctors, nurses, psychologist, psychiatrist, counsellors, social worker and peer educators. The clinic provides access to individualised regimens for DR-TB patients based on DST through ambulatory care (Fig 1).

*Eligibility for imipenem.* Patients not responding to standardised DR-TB treatment regimen in either public or private health facilities are referred to clinic. After clinical and bacteriological assessment, including DST for comprehensive resistance profile, eligible patients are enrolled for treatment at the clinic.

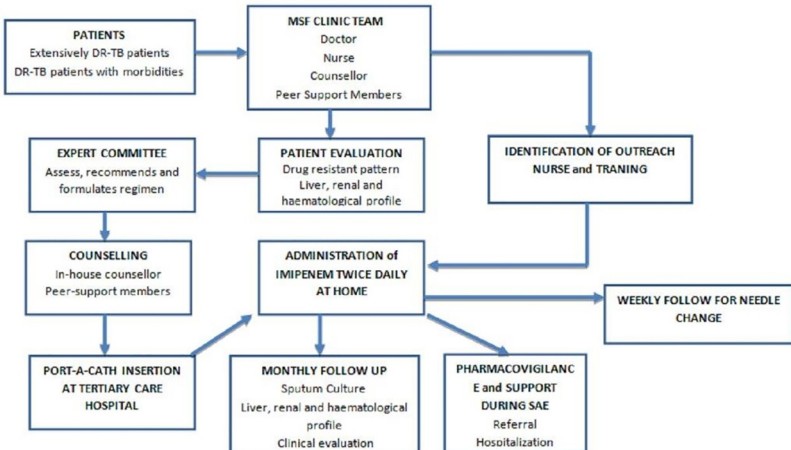

**Fig 1. The flow-diagram depicting the flow of XDR and pre XDR-TB patients initiated on imipenem containing regimen in the MSF clinic, Mumbai, India during 2015 to 2018.**

When regimen with drugs cannot be composed due to resistance, previous exposure or intolerance to other second line drugs in spite of including new and repurposed drugs Imipenem is used in patients who need a carbapenem to increase the chance of effective treatment.

*Expert committee review.* The DST and clinical profile of patients eligible for imipenem containing regimen are reviewed by expert review committee which decides the need for imipenem containing regimen with minimum 4 to 5 likely effective active drugs and mode of administration of Imipenem. Newer drugs like bedaquiline (BDQ), delamanid (DLM) and repurposed drugs like linezolid (LNZ), clofazimine (CFZ) are usually part of the regimen.

*Administration of imipenem.* The port-a-cath is inserted at tertiary care centre as a day-care procedure. Once the site is intact, without any sign of leakage, dislocation and infection, first dose of Imipenem (1 gram: average adult dose) is administered at the clinic. Imipenem is always co-administered with oral amoxicillin-clavulanate (500mg-125mg) twice daily.

Subsequent doses are administered through home based care, by trained nurse practitioner identified near the patient's home. Twice daily Imipenem dose lasting for 45 minutes each are administered every day at the patient residence by nurse practitioner who's also in charge to monitor the infusion.

*Patient support.* At MSF independent clinic, counsellors provide information to patients and caregivers on treatment and port-a-cath care. Educational sessions take place prior to treatment initiation and during the regular weekly visits needed to replace the port-a-cath needle. Patients are referred to peer support groups comprising of patients receiving and completed treatment with imipenem containing regimen for social support. The clinic provides nutritional support to patient families with dry ration delivered every month for a period of one year. Patients who need hospitalization during the treatment receive regular follow up at the inpatient facility by MSF doctor, who collaborates with the hospital physicians. MSF covers hospitalisation costs, thus care provision remains free of charge for patients during the whole duration of the DRTB treatment.

*Follow-up and adverse events.* Adverse events of medical interest with Imipenem are documented in case records. Serious and non-serious adverse events are recorded during clinical follow up and relevant investigations are done. QT interval was calculated by Fredricia method and called prolonged when the interval was more than 500 milliseconds. Similarly, arrhythmias were considered as severe cardiac event. Non-serious adverse events are managed at the clinic. Serious adverse events are recorded as a part of pharmacovigilance and relevant

corrective actions like drug withhold, drug stoppage and hospitalization are undertaken. Port-a-cath site local infection are noted, relevant samples are sent from local site and blood from opposite arm for culture to ascertain presence of septicaemia. Adverse events due to port-a-cath device, blockage, suture rupture, device dislodgement, accidental injury to the port-a-cath site are referred to tertiary care centres for further management. Sputum samples for AFB cultures and blood investigations are done every month during their treatment.

Treatment is usually administered for a total duration of at least 18–20 months, with Imipenem usually provided for minimum six months. Key factors that can influence clinical decision of Imipenem duration are time to culture conversion, number of remaining drugs likely to be effective if Imipenem is stopped, toxicity and incurrence of adverse events, Imipenem or port-a-cath related. In cases of extra pulmonary tuberculosis, response to treatment is assessed by relevant radiological and clinical investigations. The treatment outcomes adapted from the national guidelines is given in Table 1 [14].

## Study population

**Quantitative.** We included all the XDR-TB and pre-XDR TB patients initiated on imipenem containing regimen between January 2015 and June 2018 at MSF independent clinic at Mumbai.

**Qualitative.** Patients initiated on Imipenem (n = 5) and health care providers (n = 7) involved in managing patients receiving imipenem at MSF independent clinic were included. Purposive sampling technique was used to identify patients and healthcare providers for the study. Maximum variation purposive sampling was adapted to select patients (based on occurrence of adverse events and treatment outcomes) and the homogeneity sampling (those involved in administration of treatment through port-a-cath) for selection of healthcare providers [15]. The final sample size was guided by the saturation of findings.

## Data variables, sources of data and data collection

**Quantitative.** Data on demographic variables like age, gender, date of initiation of imipenem containing regimen, reason for starting imipenem regimen, site of TB, history of TB

**Table 1. Treatment outcomes among XDR and pre-XDR TB patients initiated on imipenem containing regimen.**

| Treatment Outcomes | Definition |
|---|---|
| Cured | Treatment completed as recommended by the National Policy without evidence of failure and three or more consecutive culture taken at least thirty days apart during continuation phase are negative including culture at end of treatment. |
| Treatment completed | Treatment completed as recommended by the national policy without evidence of failure but no record that three or more consecutive cultures taken at least thirty days apart are negative after intensive phase. |
| Culture conversion | Patient is considered to have culture converted when two consecutive cultures taken at least thirty days apart are found to be negative. In such case, the specimen collection date of the first negative culture is used as date of conversion. |
| Failure | Treatment terminated or need for permanent regimen change of at least two or more drugs in CP because of lack of microbiological conversion by the end of the extended intensive phase or microbiological reversion in the continuation phase after conversion to negative or evidence of additional acquired resistance. |
| Lost to follow up (LFU) | A TB patient whose treatment was interrupted for one consecutive month or more |
| Not evaluated | A TB patient for whom no treatment outcome is assigned; this includes former 'transfer-out' patients |
| Died | A patient who has died during the course of anti-TB treatment |
| Still on treatment but culture negative | A patient who has not yet completed his treatment but has culture converted |

treatment, Diabetes status, HIV status, hepatitis B status, hepatitis C status, weight in kilo-grams, height in centimetre, resistance pattern, drugs used in regimen, adverse drug reactions, port-a-cath related complications, episodes of hospitalization, date of stopping imipenem, sputum conversion status, date of sputum conversion, treatment outcomes were extracted from the patient records. The principal investigator extracted data during month of June, 2019.

**Qualitative.** Face-to-Face interviews were conducted using an open-ended interview guide. The interview guides were translated in local language (Hindi). Separate interview guides were used to interview health care providers and patients. Interviews were conducted by co-investigator, who is a female researcher (MPH), fluent in local language (Hindi) and trained in qualitative research and is not directly involved in patient care. Interviews were carried out in the preferred language of the participants (English or Hindi). All the interviews with healthcare providers were conducted in their workplace in a separate cabin to maintain privacy and confidentiality. The interviews with patients were conducted in MSF independent clinic. The identified eligible participants were explained the purpose of the study, study objectives, and process of interviews (including consent for audio-recording) in detail by the interviewer with support of the healthcare provider. If the participants agreed, informed consent was taken from the participants. During the time of interview, the interviewer repeated about the data confidentiality and safety procedures, and answered all the queries of the participant before the start of the interview. After obtaining informed consent the interviews were audio recorded and lasted for on average 28 (range: 8–63) mins. At the end of each interview, the interviewer shared the summary of findings for participant validation.

## Data entry and analysis

**Quantitative.** Data was double-entered and validated using EpiData 3.1 software (Epi-Data Association, Odense, Denmark). Demographic characteristics, clinical characteristics, comorbid conditions and baseline drug sensitivity pattern were summarised. The drugs used in the regimen, combination of drugs and imipenem use for more than six months were described with frequency and percentage. The individual adverse drug events were summarized as percentages. The incident rates of port-a-cath block, port-a-cath infection and hospitalization were calculated and expressed as episodes per 100 person-months (PM). The 'cured', 'treatment completed' and 'still on treatment with sputum conversion' were combined as 'favourable' treatment outcomes. Sputum conversion and favourable treatment outcomes were summarized as proportion with 95% confidence interval. The time to sputum conversion was calculated and summarized as median with interquartile range.

**Qualitative.** We used thematic network analysis framework, as described by Attride-Stirling [16]. The thematic network facilitated the structuring of textual data and depiction of main themes in one nexus. The interviewer prepared the transcripts within two days of conducting the interview. Manual descriptive content analysis of the transcripts was done by the VC and PT to identify the codes. The codes identified, were then grouped in categories or sub-themes and thereafter as themes. The decision on the final coding and theme generation was done by using standard procedures and in consensus. The findings were reported by using 'Consolidated Criteria for Reporting Qualitative Research' (COREQ guideline) [12].

## Ethics

The study proposal was approved by Ethics Advisory Group of the International Union Against Tuberculosis and Lung Disease, Paris, France (132/18), Ethics Review Board of Méde-cins Sans Frontières, Geneva, Switzerland and Institute Ethics Committee of Jupiter Hospital,

Thane (20/2/19). Informed written consent was taken from study participants included in the interview.

## Results

### Quantitative

We included all 70 patients initiated on imipenem through port-a-cath. The mean age of the patients was 28.1 (standard deviation: 11.2) and 36 (51.4%) were females. In total all 70 (100%) were resistant to fluoroquinolone group, 51 (72.9%) had XDR-TB, 65 (92.9%) were on retreatment after failure of second line anti tubercular drug treatment and 63 (90.0%) had pulmonary TB. The resistance to kanamycin and capreomycin was seen in 47 (67.1%) and 42 (60.0%) respectively. Of those with pulmonary TB, 36 (57.1%) had positive sputum culture during initiation of imipenem (Table 2).

Bedaquiline and delaminid were prescribed in 52 (74.3%) and 68 (97.1%) patients respectively. The combination with imipenem, bedaquiline, delaminid, linezolid and clofazimine was used in 43 (61.4%) patients (Table 3). The median duration of imipenem administration was 264 (inter quartile range: 177–367) days.

Adverse events of medical interest during treatment with imipenem are described in Table 4. Vomiting was reported by 55 (78.6%) and at least one episode of QTC prolongation more than 500 msec calculated by Fridercia method on electrocardiogram was seen in 25 (35.7%) patients. At least one episode of port-a-cath block and infection was seen in 11 (15.7%) and 20 (28.6%) respectively. At least one episode of hospitalization was seen in 23 (32.9%) patients. The rate of port-a-cath block was 2.9 per 100 PM and rate of infection was 3.5 per 100 PM. The rate of hospitalization was 5.0 per 100 PM.

Among 36 patients with positive sputum culture at initiation of imipenem, 26 (72.2%) had sputum culture conversion. The median duration between imipenem initiation and sputum conversion was 75 (inter quartile range: 60 to 110) days.

Favourable treatment outcomes were seen in 43 (61.4%, 95% CI- 49.0%-72.8%) patients. Of the 70 patients, 22 (31.4%) died, 2 (2.9%) were lost to follow-up and 3 (4.3%) were declared as treatment failure. Those treated with combination bedaquiline, delaminid, linezolid and clofazamine had favourable treatment outcome rate of 65.2% whereas those on imipenem containing regimen with either bedaquiline or delaminid with other drug combinations had 55.6% (Table 5).

### Qualitative

**Participant characteristics.**   Five patients were included in the interviews. The age of the patients ranged between 18–43 years, and three were females. Three patients had completed the treatment while two patients were on treatment at the time of interview. The two patients 'on-treatment' had completed more than 12 months of treatment. Among the healthcare providers (n = 7), one doctor, two counsellors and four nurses were interviewed.

### Challenges in delivering imipenem through port-a-cath in ambulatory patients

The overarching theme of the qualitative analysis was: Challenges in delivering Imipenem via port-a-cath device in ambulatory care. The over-arching theme consisted of four sub-themes: 1) Patient related challenges, 2) Imipenem related challenges, 3) Port-a-cath related challenges and 4) Home based care related. A total of 26 codes were categorized under these sub-themes, as summarized in Fig 2.

**Table 2. Demographic, clinical and drug sensitivity pattern of XDR-TB and pre XDR-TB patients initiated on Imipenem containing regimen between January 2015 and June 2018 at MSF clinic in Mumbai, N = 70.**

| Characteristics | Categories | Frequency | (%) |
|---|---|---|---|
| **Age (in years)** | 0–14 | 3 | (4.3) |
| | 15–24 | 31 | (44.3) |
| | 25–34 | 23 | (32.9) |
| | 35–44 | 5 | (7.1) |
| | 45–54 | 5 | (7.1) |
| | 55–64 | 3 | (4.3) |
| **Gender** | Male | 36 | (51.4) |
| | Female | 34 | (48.6) |
| **Type of TB** | New | 3 | (4.3) |
| | Retreatment after LTFU | 1 | (1.4) |
| | Retreatment after failure | 65 | (92.9) |
| | Relapse# | 1 | (1.4) |
| **Site of TB** | Pulmonary TB | 63 | (90.0) |
| | Extra-pulmonary TB | 7 | (10.0) |
| **Culture at initiation of Imipenem (n = 63)** | Positive | 36 | (57.1) |
| | Negative | 27 | (42.9) |
| **Diabetes** | Yes | 4 | (5.7) |
| | No | 66 | (94.3) |
| **Hepatitis C** | Yes | 2 | (2.9) |
| | No | 68 | (97.1) |
| **HIV** | Yes | 1 | (1.4) |
| | No | 69 | (98.6) |
| **Hepatitis B** | Yes | 1 | (1.4) |
| | No | 69 | (98.6) |
| **Resistance Pattern*** | Isoniazid | 70 | (100.0) |
| | Rifampicin | 70 | (100.0) |
| | Ethambutol | 64 | (91.8) |
| | Pyrazinamide | 66 | (94.3) |
| | Levofloxacin | 70 | (100.0) |
| | Moxifloxacin | 64 | (91.8) |
| | Kanamycin | 47 | (67.1) |
| | Capreomycin | 42 | (60.0) |
| | Ethionamide | 66 | (94.3) |
| | P-Amino Salicylic acid (PAS) | 36 | (51.4) |

* Multiple options are possible;

# Relapse- A patient who has been diagnosed with tuberculosis after s/he was declared cured or treatment completed

Abbreviation: DR-TB- Drug Resistant Tuberculosis; DST- Drug Sensitivity Testing; MTB- Mycobacterium Tuberculosis; Rif- Rifampicin; HIV- Human Immuno-deficiency Virus.

**Patient related.** The initial refusal to imipenem containing regimen by patients was reported by providers and patients themselves. This necessitated repeated counselling by the counsellors and peer support group members.

*"I was crying so much.. that I don't want it. . . what is it.. but then everyone made me understand.. Then only I said yes and it was started.."*

- (43 year female patient)

**Table 3. Drugs used in the regimen and duration of imipenem in management of XDR-TB and pre XDR-TB patients initiated on Imipenem containing regimen between January 2015 and June 2018 at MSF clinic in Mumbai, N = 70.**

| Characteristics | Categories | Frequency | Percentage |
|---|---|---|---|
| **Drugs used in regimen** | Bedaquiline | 52 | (74.3) |
| | Delaminid | 68 | (97.1) |
| | Amoxicillin-Clavulanate | 70 | (100.0) |
| | Linezolid | 64 | (91.4) |
| | Clofazimine | 65 | (92.9) |
| | Cycloserine | 30 | (42.9) |
| | Moxifloxacin | 36 | (51.4) |
| | Ethionamide | 17 | (24.3) |
| | P-Amino Salicylic acid (PAS) | 33 | (47.1) |
| **Drugs combination with imipenem** | Bedaquline + Delaminid + Linezolid + Clofazimine | 43 | (61.4) |
| | Others | 27 | (38.6) |
| **Duration of Imipenem** | ≤ 6 months | 19 | (27.1) |
| | > 6 months | 51 | (72.9) |

Abbreviation: DR-TB- Drug Resistant Tuberculosis.

**Table 4. Drugs and port-a-cath related adverse events among XDR-TB and pre XDR-TB patients initiated on imipenem containing regimen between January 2015 and June 2018 at MSF clinic in Mumbai, N = 70.**

| Characteristics | Categories | n | (%) |
|---|---|---|---|
| **Adverse Events (AEs)** | Nausea/Vomiting | 55 | (78.6) |
| | Diarrhea | 14 | (20.0) |
| | Pruritus | 2 | (2.9) |
| | Rash | 9 | (12.9) |
| | Thrombocytosis | 1 | (1.4) |
| | Acute renal failure | 7 | (10.0) |
| | Seizure | 4 | (5.7) |
| | QTC prolongation (>500 msec) | 25 | (35.7) |
| | Septicemia | 8 | (11.4) |
| | Hepatic toxicity | 7 | (10.0) |
| **Port-a-cath related complications** | | | |
| **Episodes of Blockage** | No block | 59 | (84.3) |
| | 1 | 7 | (10.0) |
| | 2 | 1 | (1.4) |
| | 3 | 3 | (4.3) |
| **Episodes of Infection** | No infection | 50 | (71.4) |
| | 1 | 18 | (25.7) |
| | 2 | 2 | (2.9) |
| **Episodes of hospitalization** | No hospitalization | 47 | (67.1) |
| | 1 | 17 | (24.3) |
| | 2 | 4 | (5.7) |
| | 3 | 2 | (2.9) |

Abbreviation: DR-TB- Drug Resistant Tuberculosis.

**Table 5. Treatment outcomes stratified by combination of drugs among XDR-TB and pre XDR-TB patients initiated on Imipenem containing regimen between January 2015 and June 2018 at MSF clinic in Mumbai, N = 70.**

| Treatment Outcomes | IBDLC, N = 43 | Other imipenem containing regimen, N = 27 | Total, N = 70 |
|---|---|---|---|
| | n (%) | n (%) | n (%) |
| **Favourable Outcome** | **28 (65.2)** | **15 (55.6)** | **43 (61.4)** |
| Cured | 16 (37.2) | 13 (48.1) | 29 (41.4) |
| Treatment Completed | 8 (18.6) | 2 (7.5) | 10 (14.3) |
| On treatment culture negative (favourable outcome) | 4 (9.4) | 0 (0.0) | 4 (5.7) |
| **Unfavourable Outcome** | **15 (34.8)** | **12 (44.4)** | **27 (38.6)** |
| Died | 11 (25.6) | 11 (40.7) | 22 (31.4) |
| Loss to follow-up | 2 (4.6) | 0 (0.0) | 2 (2.9) |
| Failure | 2 (4.6) | 1 (3.7) | 3 (4.3) |

Abbreviation: IBDLC- Imipenem + Bedaquiline + Delaminid + Linezolid + Clofazamine

Other Imipenem containing regimen–regimen with either Bedaquiline or Delaminid with effective second line drugs.

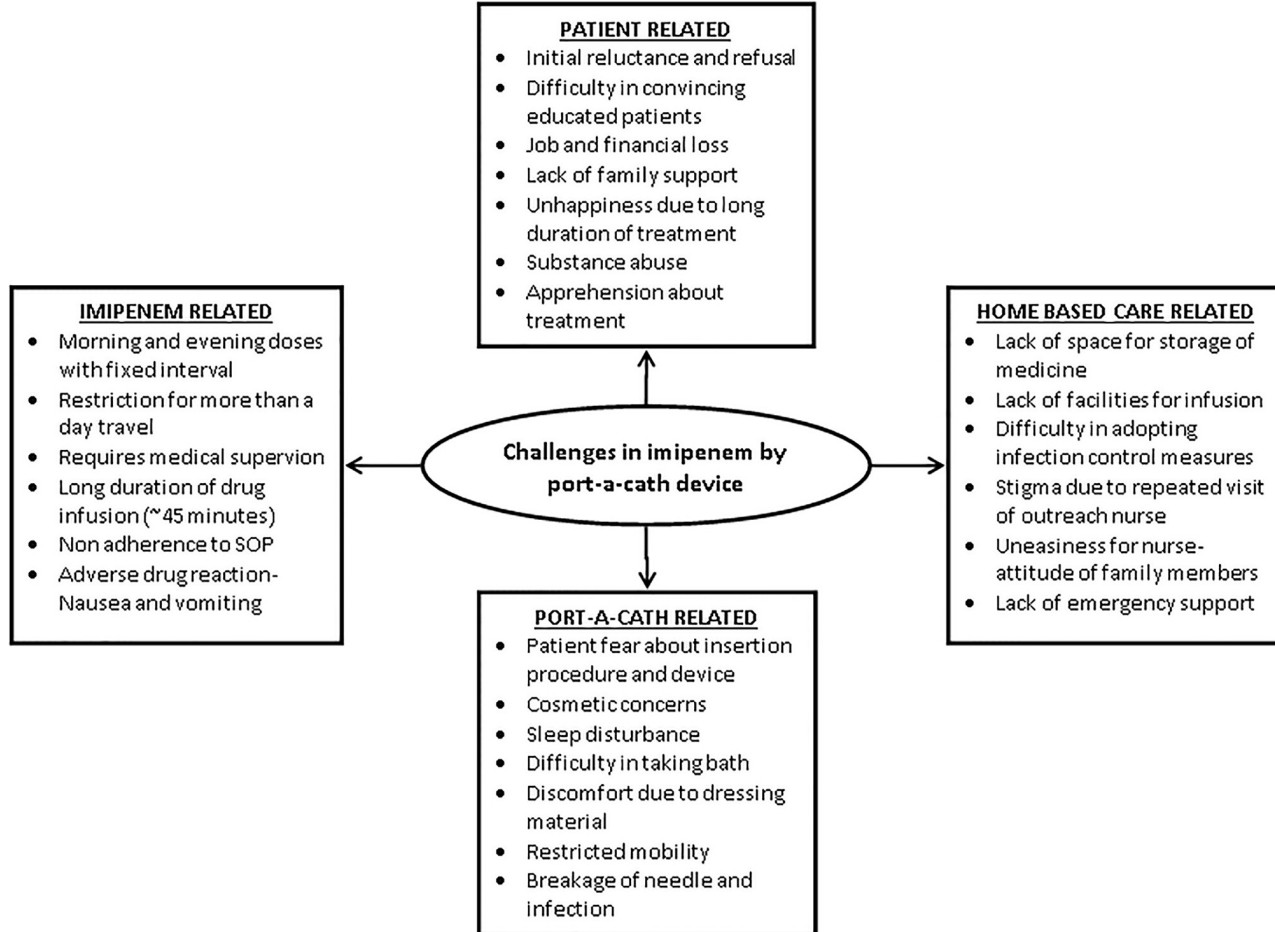

**Fig 2. Challenges in provision of imipenem through port-a-cath for XDR and pre XDR-TB patients as perceived by patients (n = 5) and health care providers (n = 7) at MSF clinic, Mumbai during 2015 to 2018.**

The healthcare providers felt, it was difficult to counsel educated patients as they used to assess the internet for complications and deny the treatment.

*"Majority of these patients were educated.. they were not someone who would just listen to you if you say something.. one was engineer, other was banker.. people google a lot .. they get to know about PORT infection and then they come and discuss with us"*

- Counsellor

The healthcare provider's narrated instances where patients lost their existing job and were emotionally disturbed because of the treatment.

*"He [patient] is the breadwinner of the family. He took leaves from office 1–2 months, and he was doing some work from office. Due to Imp he could not go to work, now his position is getting replaced. Losing a job is very stressful for him. Emotionally this is a struggle.."*

- Outreach Nurse

The healthcare providers felt patients were unhappy with injectable drugs and kept requesting them to stop the same.

*"Patient count days for imipenem . . . They always ask please see my report.. 'how long this [imipenem injection] will continue'."*

- Counsellor

**Imipenem related.**   As daily two doses of imipenem were to be delivered at a fixed interval of 12 hours, the patients and families had to adjust their lifestyles to accommodate the same. Patients had to come back to home whenever it was time for injection.

*"It was difficult for us to accept this alteration in our lifestyle because we couldn't go out. We had to be back within 12 hours to give Imp next dose. He can't administer it for himself. It had become kind of setback for us, also for him."*

- Outreach nurse providing injections to her father

The patients expressed grief that they were not able to follow their social and moral obligations due to the treatment which required them to stay in home.

*"My grandfather died but I didn't go to my village."*

- 19 year old male patient

The drug had to be infused over 45 minutes and patients reported having palpitations if drug is infused fast. The patients and counsellors expressed displeasure that the outreach nurses didn't adhere to SOPs and tried to finish the infusion fast.

*"Once there was a new nurse, she gave the injection to me in 25 min. This was supposed to be given for 45 min. My heart beat raised.."*

- 19 year old male patient

All the interviewees reported that the patients had vomiting with imipenem. The patients also reported difficulty in consuming food because of consistent vomiting and bad taste.

*"I had vomiting.. I could not eat food.. When they used to flush at the end of the dose, the medicine taste used to go in the mouth."*

- 23 year old female patient

**Port-a-cath related.** The healthcare providers and patients mentioned that they were anxious about the procedure for inserting port-a-cath. The healthcare providers narrated that, patients had raised concerns about device and its placement over the chest.

*"The patients feel restricted. They feel something is there in their body.. Female asks us, 'it is not visible right'.. even men are conscious, they ask 'It will not rust right'.. . . few asks, 'It is placed near heart, does it not cause heart attack?'.."*

- Counsellor

Patients reported disturbed sleep as they had to avoid sleeping on the side the port-a-cath was inserted. In case patients rolled on the side with port-a-cath during sleep, then there used to be damage to the needle and had to be replaced.

*"It was difficult to sleep.. I could not sleep on the same side where it was placed."*

- 18 year old male patient

The patients also expressed displeasure that they couldn't take bath regularly with the port-a-cath.

*"I think when the port was inserted there was some blood that went into my hairs. But I was not allowed to wash my hair for a month, that was difficult time for me.. Because all the smell stayed in hair, that was difficult."*

- 23 year old female patient

The healthcare providers felt that patients repeatedly came to emergency clinics with either block or displacement of port-a-cath needle.

*"So sometimes they sleep over it or fell down, the needle gets in or gets out.. it gets blocked or displaced"*

- Doctor

To avoid such displacements and blocks, the patients had to restrict the mobility of the right hand. The healthcare providers noted that, this restriction limited their job opportunities and patients had to discontinue treatment to work.

*"He [patient] received multiple counselling sessions, he was told how he could work with this [port-a-cath], how much load he could carry.. but still he denied and left."*

- Doctor

The patients complained about discomfort with port-a-cath due to irritation due to adhesive tapes used for fixing port-a-cath.

*"I had redness and rashes at the point of sticking"*

– 19 year old male patient

**Home-based care related.** The patients felt it was difficult to find a nurse to come home and provide imipenem infusion.

*"the hospital said, 'No, we will not give' [imipenem].. Later MSF tem convinced the main hospital doctor and the nurse agreed"*

- 19 year old male patient

The healthcare providers and patients reported that there was difficulty in storing the medicines due to space constraints in the patient's house.

*"Storage is an issue.. when we explain along with medicine, other equipments, NS needs to be kept at your house.. they complain of not much space at home."*

- Counsellor

The healthcare providers complained that, it was very difficult to maintain the sterility in the house and this lead to repeated infections at port-a-cath site.

*"main difficulty was to maintain sterility at home. When you are in a hospital setup, you have everything according to Infection control policies but at home it is a very different setup"*

- Outreach nurse

The healthcare providers felt there was lack of facilities to administer imipenem and several modifications had to be done in the house.

*"I used to take the medicine on the bed, the drip was hanged on the hook for curtains"*

- 19 year old male

The healthcare providers informed that the patients were concerned about nurse visiting their house due to stigma. This was more so, when the patient was young female.

*"More issues with families of female patients. . . they [patients] say neighbours may say something as we visit every day.. they request us not to disclose this to neighbours as it may cause difficulty in her marriage"*

- Outreach nurse

## Discussion

We conducted a mixed-methods study to assess adverse events, treatment outcomes and challenges in delivering ambulatory care for XDR-TB and pre XDR-TB patients on imipenem through port-a-cath. Our study had several key findings. First, sputum conversion was seen in 72% of cases and 61% had favourable treatment outcomes. Second, vomiting was the most common adverse event followed by QTC prolongation. Third, 28% cases had minimum one episode of port-a-cath infection and 16% had minimum one episode of port-a-cath block. Fourth, the challenges in using Imipenem as a part of the treatment regimen included patient's difficulties in adhering to timelines, vomiting, restricted mobility due to port-a-cath, stigma due to repeated house visits of nurses, lack of space for drug storage and constraints for infection control at patient's home.

Similar to our study, a multicentre study conducted in Europe and Southern American countries had reported 72% of sputum conversion rate among DR-TB patients.[9] About 32% of patients with previous study had MDR-TB, whereas our study had patients with more complicated resistance pattern. The previous study had relatively lower treatment success rate (60%) in spite of describing a cohort of patients having less complicated resistance pattern. The high favourable outcome rate in our study might be due one or more of the listed reasons. 1) Concomitant use of both newer drugs (bedaquiline and delaminid) in about 70% of the patients for the entire duration of treatment. 2) The patients in the study cohort received social, financial and nutritional support during the treatment. 3) Port-a-cath was used for delivering the drug, thus reducing the pain caused by repeated intravenous injections. 4) Ambulatory care with nurses providing home based care to patients. However, the treatment outcomes in the current cohort was poor compared to Nix-TB trial with all oral shorter regimen having a favourable outcomes among 95 out 107 XDR-TB patients [17].

Vomiting was the most common adverse events (AEs) noted similar to other studies [8,9,18,19]. QTC prolongation was second most common adverse event seen. QT interval correction was calculated by Fridericia method and QT interval above 500 msec was considered as adverse event. QT interval prolongation is common with electrolyte abnormalities which could be due to vomiting, poor nutritional intake, along with the QTC prolonging drugs like bedaquiline, delaminid and Clofazimine used in the regimen [20–22]. However, no severe cardiac event especially arrhythmias were reported. Port-a-cath related complications were relatively common and more frequent than previously reported among cohorts of patients on chemotherapy [9,23–27]. High rates of port-a-cath infections may be due to fact that most of the patients were residing in urban slums with poor infection control facilities. Also, the use of the port-a-cath twice a day on daily basis likely increases chances of infections.

Mortality though remained high in our cohort, with 31% of reported deaths during treatment, extensive disease, late diagnosis and late referral to MSF clinic might be the contributory factors. The systematic review on carbapenems concurred with our findings of effectiveness of carbapenems and safety and tolerability being good, with the proportion of adverse events attributable to carbapenems below 15% [28].

The study has several strengths. First, this is the first study as per our knowledge to have assessed the treatment outcomes of regimens including imipenem delivered through port-a-cath on ambulatory basis. Second, the study was conducted in the programmatic setting of MSF independent clinic and reflects the ground realities. Third, selection bias was negated by including all the patients on treatment during the study reference period. Fourth, we used mixed methods study design which provided insights on the challenges faced by patients and healthcare providers in delivering imipenem through port-a-cath. Fifth, we adhered to STrengthening the Reporting of OBservational studies in Epidemiology (STROBE) and COREQ guidelines for reporting the study findings.

The study has a few limitations. First, due to deficiencies in data source, we failed to collect details on drug exposure history and duration, number of previous TB episodes, duration between eligibility and initiation of Imipenem which could have provided better insights to interpret the favourable treatment outcome rates. Second, adverse events (AE) details were extracted from case records. Hence, we might have underestimated the proportion and rates of adverse events due under reporting of the same. Also, we failed to relate the AEs with specific drug in the regimen. Third, reasons for lost-to-follow-up and causes of death among the patients were not available. Fourth, the study results may not be generalizable as this is a unique health facility providing social, financial and nutritional support to patients. Thus, the favourable treatment outcome rate might be higher than routine programme settings. Sixth, we don't have any existing literature on similar use of port-a-cath in DRTB treatment to

compare and contrast our findings. Lastly, the favourable outcomes cannot be attributed to imipenem alone, since DR-TB treatment involved combination of active drugs.

The study findings have some implications for management of XDR-TB and pre XDR-TB patients.

The favourable outcomes among XDR-TB and pre-XDR-TB patients were promising with imipenem containing regimen. According to the Annual India TB 2019 report, XDR-TB patients started on treatment under National Tuberculosis Programmes (NTP) had favourable outcomes only in 27% patients compared to 61% in our cohort and mortality in 42% patients compared to 32% in our cohort [29]. Although, all oral drug regimens are preferred, NTPs can consider imipenem when an effective regimen can't be designed with only oral drugs. The early initiation of individualized regimens based on DST results can yield better treatment outcomes. Therefore, the DST for newer and repurposed drugs has to be made available to objectively formulate an effective regimen.

More than three-fourths of the patients reported vomiting with imipenem. Severe complications like seizure, acute renal failure and hepatotoxicity were also reported. Pharmacovigilance to monitor the safety of this regimen and appropriate management of adverse events are recommended.

It is feasible to deliver imipenem through port-a-cath which prevents repeated injections and hospitalization. NTP can consider port-a-cath as an option to deliver imipenem. However, existing facilities have to be strengthened for port-a-cath insertion procedures, weekly needle change and management of the port-a-cath related adverse events. Similarly, there is need for testing feasibility, acceptance and effectiveness of other intravenous dwelling catheters like peripherally inserted central catheter (PICC) for delivering intravenous infusions.

Job loss, stigma, apprehension about treatment outcomes, restriction for travel and socialization have detrimental effect on quality of patient's life [30]. The capacity building of existing counsellors or medical social workers in the NTPs for emotional and psychological support for patients through counselling has to be considered. The peer support forums have to be established and patients have to be linked to such forums. Nutritional support can be made available to potentially avert severe adverse drug events and improve the innate immunity. Vocational training could be offered to make the patient productive in spite of his physical limitations and overcome the feeling of worthlessness.

Although home based treatment was possible, both patients and healthcare providers faced several challenges. There is need for exploring other options for ambulatory care. The patients can be requested to visit nearby day care facilities, receive the infusions and return back to house. However, support for travel has to be made available if such options are explored.

## Conclusion

The treatment outcomes with imipenem containing regimen on ambulatory care was promising with significant majority having favourable outcome. Despite the several challenges in delivering care at home, administration of Imipenem was feasible with port-a-cath. Programmes need to explore alternative options like day care facilities for management of patients on ambulatory basis.

## Acknowledgments

This research was conducted through the Structured Operational Research and Training Initiative (SORT IT), a global partnership led by the Special Programme for Research and Training in Tropical Diseases at the World Health Organization (WHO/TDR). The model is based on a course developed jointly by the International Union Against Tuberculosis and Lung

Disease (The Union) and Medécins sans Frontières (MSF/Doctors Without Borders). The specific SORT IT programme which resulted in this publication was jointly developed and implemented by: The Union South-East Asia Office, New Delhi, India; the Centre for Operational Research, The Union, Paris, France; Medécins sans Frontières (MSF/Doctors Without Borders), India; Department of Preventive and Social Medicine, Jawaharlal Institute of Postgraduate Medical Education and Research, Puducherry, India; Department of Community Medicine, All India Institute of Medical Sciences, Nagpur, India; Department of Community Medicine, ESIC Medical College and PGIMSR, Bengaluru, India; Department of Community Medicine, Sri Manakula Vinayagar Medical College and Hospital, Puducherry, India; Karuna Trust, Bangalore, India; Public Health Foundation of India, Gurgaon, India; The INCLEN Trust International, New Delhi, India; Indian Council of Medical Research (ICMR), Department of Health Research, Ministry of Health and Family Welfare, New Delhi, India; Department of Community Medicine, Sri Devraj Urs Medical College, Kolar, India; and Department of Community Medicine, Yenepoya Medical College, Mangalore, India.

## Author Contributions

**Conceptualization:** Vijay Vinayak Chavan, Sharath Nagaraja.

**Data curation:** Vijay Vinayak Chavan, Roma Paryani, Pramila Singh, Mrinalini Das, Petros Isaakidis.

**Formal analysis:** Vijay Vinayak Chavan, Alpa Dalal, Pruthu Thekkur, Homa Mansoor, Augusto Meneguim, Mrinalini Das, Gabriella Ferlazzo, Petros Isaakidis.

**Investigation:** Vijay Vinayak Chavan, Sharath Nagaraja, Gabriella Ferlazzo.

**Methodology:** Vijay Vinayak Chavan, Sharath Nagaraja, Pruthu Thekkur, Stobdan Kalon, Mrinalini Das, Gabriella Ferlazzo, Petros Isaakidis.

**Project administration:** Homa Mansoor, Augusto Meneguim, Stobdan Kalon, Mrinalini Das, Gabriella Ferlazzo, Petros Isaakidis.

**Resources:** Pruthu Thekkur, Augusto Meneguim, Roma Paryani, Pramila Singh, Stobdan Kalon, Mrinalini Das, Gabriella Ferlazzo.

**Software:** Pruthu Thekkur.

**Supervision:** Alpa Dalal, Sharath Nagaraja, Pruthu Thekkur, Homa Mansoor, Augusto Meneguim, Pramila Singh, Stobdan Kalon, Mrinalini Das, Gabriella Ferlazzo, Petros Isaakidis.

**Validation:** Vijay Vinayak Chavan, Sharath Nagaraja, Pruthu Thekkur, Augusto Meneguim, Roma Paryani, Stobdan Kalon, Mrinalini Das, Gabriella Ferlazzo, Petros Isaakidis.

**Visualization:** Vijay Vinayak Chavan, Pruthu Thekkur, Petros Isaakidis.

**Writing – original draft:** Vijay Vinayak Chavan, Sharath Nagaraja, Pruthu Thekkur, Homa Mansoor, Augusto Meneguim, Mrinalini Das, Petros Isaakidis.

**Writing – review & editing:** Vijay Vinayak Chavan, Alpa Dalal, Sharath Nagaraja, Pruthu Thekkur, Homa Mansoor, Augusto Meneguim, Roma Paryani, Pramila Singh, Stobdan Kalon, Mrinalini Das, Gabriella Ferlazzo, Petros Isaakidis.

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
