## [Decision Letter · Decision Letter 0]

10 Dec 2019

PONE-D-19-25948

Ambulatory management of pre and extensively drug resistant tuberculosis patients with Imipenem delivered through port-a-cath : A mixed methods study on treatment outcomes and challenges.

PLOS ONE

Dear dr chavan,

Thank you for submitting your manuscript to PLOS ONE. After careful consideration, we feel that it has merit but does not fully meet PLOS ONE’s publication criteria as it currently stands. Therefore, we invite you to submit a revised version of the manuscript that addresses the points raised during the review process.

Please remove all the data of the qualitative study, to be published in a different paper, and stick only to the quantitive data, as explained by the reviewer in this case.

We would appreciate receiving your revised manuscript by Jan 24 2020 11:59PM. To enhance the reproducibility of your results, we recommend that if applicable you deposit your laboratory protocols in protocols.io, where a protocol can be assigned its own identifier (DOI) such that it can be cited independently in the future. For instructions see: http://journals.plos.org/plosone/s/submission-guidelines#loc-laboratory-protocols

We look forward to receiving your revised manuscript.

Kind regards,

Pere-Joan Cardona, MD, PhD

Academic Editor

PLOS ONE

Journal Requirements:

3. Your ethics statement must appear in the Methods section of your manuscript. If your ethics statement is written in any section besides the Methods, please move it to the Methods section and delete it from any other section. Please also ensure that your ethics statement is included in your manuscript, as the ethics section of your online submission will not be published alongside your manuscript.

Reviewers' comments:

Reviewer's Responses to Questions

**Comments to the Author**

1. Is the manuscript technically sound, and do the data support the conclusions?

Reviewer #1: Yes

2. Has the statistical analysis been performed appropriately and rigorously? 

Reviewer #1: Yes

3. Have the authors made all data underlying the findings in their manuscript fully available?

Reviewer #1: No

4. Is the manuscript presented in an intelligible fashion and written in standard English?

Reviewer #1: Yes

5. Review Comments to the Author

Reviewer #1: Comments for the Authors

General:

It was a pleasure to review your manuscript. You have performed a great job in the field and high quality research. My main concerns regarding the manuscript, in general, is that you are trying to present in the same manuscript two what is seems to me, two different studies using mixed methods study (one quali and other quantitative). It lengthens the text and causes it to be in some unstructured and confusing parts. I would suggest splitting both parts and try to publish qualitative and quantitative parts separately. Other concerns are that the introduction and discussion sections need to state more clearly the priority of implementation of new guidelines of M/XDR TB management launched by WHO using new drugs and regimens leaving injectables and iv treatment use as a secondary or tertiary alternatives (E.g.: new regimens with Linezolid, Bedaquiline, Delamanid, Pretomanid will probably could bring better treatment outcomes with less adverse events if possible to use. Introduction section is too long and discussion too short and needs more work and order. Some parts of the analysis should be reworked and some references need to be up dated. Please, also note that even though patients and probably data were collected in a prospective manner, this is not a cohort study as it is reflected in the data analysis. If you would like to consider a cohort study, I would suggest changing analysis trying to incorporate the variable time and calculate Hazard or Risk Ratios.

Saying that I would like to underline that operational research by NGOS such is MSF in high burden M/XDR TB countries is essential as it is well demonstrated in the study. Even is not the aim of the study, it is also a cost-effective strategy treating and following in a very good way a high number of patients. After addressing the main comments and the following minor issues, the manuscript could be ready to be published.

Specific comments:

Title: according with my general comment, I’d suggest to delete the second part of the title “…a mixed methods study on treatment outcomes and challenges”.

Keywords: should be MeSH Terms in alphabetic order.

Abstract:

Please, review this carefully considering the general comments.

Introduction:

Well written and structured but too long. Try to shorten to 1 page. Up-date data and references number 1, 4, 5.

Line 102: add “…cases due to safety of second line drugs and to…”

Lines 110-115: see and comment new WHO recommendations. This paragraph is not refereeing to the new recommendations.

Redefine the objective: delete objective number 2 and add in objective number 2: “…and compare new drugs +imipenem with other no new drugs with imipenem…”

Methods:

Well-structured but seven pages are too much. Please, delete the qualitative part and try to shorten this section.

It seems that this is an observational analytical cross-sectional study instead of cohort study.

Setting: Is MSF covering all territory? Try to define the population/area of coverage. Are patients studied representative of the total/general population? Are cases representatives of the general population? This is a key issue for the external validity of the study. Please discuss.

Figure number 1 has a very bad quality. Impossible to read.

Follow up and adverse events part: are you describing the standard of care? If so, you can refer to already published guidelines.

What definitions are you using for treatment outcomes definition? Please, add a reference to support these definitions. If the definitions are the standard accepted, refer to the readers to the published definitions should be enough.

Study population: same comment above; is all population of Bombay or any defined district?

Source and data analysis:

Include only the analyzed variables and in the same order that are stated in the results section, ideally: demographic, social, risk factors, diagnostic, treatment…

Add the comparative analysis performed new drugs + imipenem with other no new drugs with imipenem. Give the 95% CI. Delete the part about time to culture conversion. I cannot see any results in this regard.

It is better to use the median and interquartile range if the continuous variables do not follow normal distribution.

Results:

Please, add relapse definition in table 2.

Table 4: adding two more columns with the two regimens (IBDLC and other) would give very good information.

Table 5 very important: add a column with the p-values.

Discussion:

Try to put a little order following the results section.

When you talk about imipenem, are you referring always to the combination with amoxiclav? What doses?

Regarding the proportion of culture conversion (26/36 patient’s culture positive) at imipenem initiation, I think is difficult to attribute it to only imipenem action. Would it be also possible to the activity of other more active anti-TB drugs (BM, Bq, Lz,..)? Please discuss.

Lines 470-471 and Table 5: when you compare both regimens IBDLC and other imipenem regimen, was the duration the same for both? Median of duration for both regimens? Any differences?

Lines 478-479: QTc prolongation definition not here. Line 482: severe cardiac event definition?

Lot of new and good ideas discussed. Left some of them as a recommendations (eg: lines 516, 521, etc.

Commnet and compare the favorable outcome with those already published (eg: Nix TB trial).

Please, try to redefine the conclusion according to the aim of the study. From my perspective, 65% of favorable outcome is not as promising as you state in the conclusion section.

Complete the manuscript with a few of important recommendations deducted from the study would be desirable.

Best regards,

6. PLOS authors have the option to publish the peer review history of their article (what does this mean?). If published, this will include your full peer review and any attached files.

Reviewer #1: No

---

## [Author Response · Author response to Decision Letter 0]

30 Jan 2020

Dear Editor .

repy to reviewer has been incorporated for further action.

thank you.

dr vijay

---

## [Decision Letter · Decision Letter 1]

6 Apr 2020

PONE-D-19-25948R1

Ambulatory management of pre and extensively drug resistant tuberculosis patients with Imipenem delivered through port-a-cath : A mixed methods study on treatment outcomes and challenges.

PLOS ONE

Dear Dr. Chavan,

Thank you for submitting your revised manuscript to PLOS ONE. After careful consideration, we feel that it has merit but does not fully meet PLOS ONE’s publication criteria as it currently stands. Therefore, we invite you to submit a revised version of the manuscript that addresses the points raised during the review process.

**Reviewer #1 considers that you have addressed his/her comments satisfactorily. Yet, since you wished to maintain the qualitative part of the study in one single manuscript, I invited a second reviewer (now Reviewer #2) to review this specific part. As you will see below, this Reviewer raised several minor concerns that should be addressed in a further revised manuscript before it can be accepted for publication.**

We would appreciate receiving your revised manuscript by May 21 2020 11:59PM. To enhance the reproducibility of your results, we recommend that if applicable you deposit your laboratory protocols in protocols.io, where a protocol can be assigned its own identifier (DOI) such that it can be cited independently in the future. For instructions see: http://journals.plos.org/plosone/s/submission-guidelines#loc-laboratory-protocols

We look forward to receiving your revised manuscript.

Kind regards,

Olivier Neyrolles

Academic Editor

PLOS ONE

Reviewers' comments:

Reviewer's Responses to Questions

**Comments to the Author**

1. If the authors have adequately addressed your comments raised in a previous round of review and you feel that this manuscript is now acceptable for publication, you may indicate that here to bypass the “Comments to the Author” section, enter your conflict of interest statement in the “Confidential to Editor” section, and submit your "Accept" recommendation.

Reviewer #1: (No Response)

Reviewer #2: (No Response)

2. Is the manuscript technically sound, and do the data support the conclusions?

Reviewer #1: Yes

Reviewer #2: Yes

3. Has the statistical analysis been performed appropriately and rigorously? 

Reviewer #1: Yes

Reviewer #2: Yes

4. Have the authors made all data underlying the findings in their manuscript fully available?

Reviewer #1: No

Reviewer #2: Yes

5. Is the manuscript presented in an intelligible fashion and written in standard English?

Reviewer #1: Yes

Reviewer #2: Yes

6. Review Comments to the Author

Reviewer #1: (No Response)

Reviewer #2: 

General comments for the authors

This is a well-researched manuscript by using mixed method design and multiple data sources on one of a major public health problem. However, it requires some clarifications and to make more comprehensive as well as be consistent throughout the document.

Abstract

Abstract method is not complete, only mentioned design, better to include study population, sampling, data collection and analysis in short.

Line 73 “… qualitative (descriptive study…)”. Is it explorative or descriptive?

Line 76 “.51 (72.9%) had XDR-TB” --- start any sentence with words not with number, Fifty one …

Introduction

Line 127 “ … its use in management of DR-TB is not documented.” Not documented where? In the study area or at all (Globally)? Need clarification.

Need to add more similar or related studies conducted elsewhere.

Methods

Under general Setting, lines 155 – 157, mention how many public health facilities, NGOs/private clinics are providing the free care to TB patients. Also how many of these facilities are providing ambulatory management of pre and extensively drug resistant TB patients?

Study population for qualitative, lines 235 and 237, you employed purposive sampling technique for both health care providers and for patients, please specify the type of purposive sampling used for each. Lines 236 – 238, why you included only 5 patients? Were only 5 patients experienced adverse drug events during treatment and treatment outcomes? Do you think that there was saturation by those 5 patients? Moreover, please specify the patients´ status during the interview, whether they were on treatment or completed the treatment.

Data collection: line 251, were the interview guides translated to local language?

Lines 258 and 259, “...the interviews were audio recorded and lasted for on average 28 (range: 8-63) mins. The minimum time taken for the interview was too short (8 mins), were these interviews lasted for few mins ( 8, 9, 10 mins ) complete? OR incomplete? As a qualitative study, it might be difficult to explore what was expected within the short time.

What data quality assurance methods you have used for the qualitative data? (for example, building trust, small talk before formal interview, member check, debriefing, ….)

Data analysis: lines 276 and 277, “The interviewer prepared the transcripts within two days of conducting the interview” why the interviewer waited for two days? Why not did daily transcription? Moreover, please describe the whole analysis method of qualitative data (codes, categories, and themes).

Result

Line 349, there should be categories/ sub-themes between codes and themes. The same is true in the method section.

For qualitative part/ challenges: you mentioned repeatedly “The healthcare providers, the patients, or the healthcare providers and patients reported/ felt/ expressed/ mentioned … “ Do you mean all, a majority, most, or some? Please specify for the sentences before the quotes. It is obvious that a single person reported what was mentioned in the quote.

Discussion

Line 480, “ … treatment outcomes in the current cohort was poor compared to Nix-TB trial with all oral shorter regimen…”, please explain the possible reasons or justify after comparing with similar study.

As a mixed study, qualitative study findings were not discussed. Need to discuss the qualitative study findings too.

7. PLOS authors have the option to publish the peer review history of their article (what does this mean?). If published, this will include your full peer review and any attached files.

Reviewer #1: No

Reviewer #2: Yes: Berhane Megerssa Ereso (Assistant Professor, PhD candidate)

---

## [Author Response · Author response to Decision Letter 1]

21 May 2020

Dear Reviewer Berhane Megerssa Ereso ,

thank you so much for your interest in our study.

i have tried my best to answer your queries in the response letter. would be glad to answer if any further more queries arise. 

thanking you once again for making our study more apt.

Regards

Dr Vijay

---

## [Decision Letter · Decision Letter 2]

1 Jun 2020

Ambulatory management of pre and extensively drug resistant tuberculosis patients with Imipenem delivered through port-a-cath : A mixed methods study on treatment outcomes and challenges.

PONE-D-19-25948R2

Dear Dr. Chavan,

We are pleased to inform you that your manuscript has been judged scientifically suitable for publication and will be formally accepted for publication once it complies with all outstanding technical requirements.

With kind regards,

Olivier Neyrolles

Section Editor

PLOS ONE

Additional Editor Comments (optional):

Reviewers' comments:

Reviewer's Responses to Questions

**Comments to the Author**

1. If the authors have adequately addressed your comments raised in a previous round of review and you feel that this manuscript is now acceptable for publication, you may indicate that here to bypass the “Comments to the Author” section, enter your conflict of interest statement in the “Confidential to Editor” section, and submit your "Accept" recommendation.

Reviewer #2: All comments have been addressed

2. Is the manuscript technically sound, and do the data support the conclusions?

Reviewer #2: Yes

3. Has the statistical analysis been performed appropriately and rigorously? 

Reviewer #2: Yes

4. Have the authors made all data underlying the findings in their manuscript fully available?

Reviewer #2: Yes

5. Is the manuscript presented in an intelligible fashion and written in standard English?

Reviewer #2: Yes

6. Review Comments to the Author

Reviewer #2: I have reviewed this manuscript.

The authors have addressed what I have raised during the review.

Thank you so much.

7. PLOS authors have the option to publish the peer review history of their article (what does this mean?). If published, this will include your full peer review and any attached files.

Reviewer #2: Yes: Berhane Megerssa Ereso

---

## [Editor Report · Acceptance letter]

4 Jun 2020

PONE-D-19-25948R2 

Ambulatory management of pre- and extensively drug resistant tuberculosis patients with imipenem delivered through port-a-cath: A mixed methods study on treatment outcomes and challenges 

Dear Dr. Chavan:

I'm pleased to inform you that your manuscript has been deemed suitable for publication in PLOS ONE. Congratulations! Your manuscript is now with our production department. 

Kind regards, 

on behalf of

Dr. Olivier Neyrolles 

Section Editor

PLOS ONE